# Linking Puberty and the Gut Microbiome to the Pathogenesis of Neurodegenerative Disorders

**DOI:** 10.3390/microorganisms10112163

**Published:** 2022-10-31

**Authors:** Pasquale Esposito, Nafissa Ismail

**Affiliations:** 1NISE Laboratory, School of Psychology, Faculty of Social Sciences, University of Ottawa, Ottawa, ON K1N 6N5, Canada; 2Brain and Mind Research Institute, University of Ottawa, Ottawa, ON K1N 6N5, Canada

**Keywords:** adolescence, neurodegeneration, microbiota, dysbiosis, sex differences, immune system, neurodevelopment, lipopolysaccharide

## Abstract

Puberty is a critical period of development marked by the maturation of the central nervous system, immune system, and hypothalamic–pituitary–adrenal axis. Due to the maturation of these fundamental systems, this is a period of development that is particularly sensitive to stressors, increasing susceptibility to neurodevelopmental and neurodegenerative disorders later in life. The gut microbiome plays a critical role in the regulation of stress and immune responses, and gut dysbiosis has been implicated in the development of neurodevelopmental and neurodegenerative disorders. The purpose of this review is to summarize the current knowledge about puberty, neurodegeneration, and the gut microbiome. We also examine the consequences of pubertal exposure to stress and gut dysbiosis on the development of neurodevelopmental and neurodegenerative disorders. Understanding how alterations to the gut microbiome, particularly during critical periods of development (i.e., puberty), influence the pathogenesis of these disorders may allow for the development of therapeutic strategies to prevent them.

## 1. Introduction

Puberty is a period of development that is accompanied by the maturation of various fundamental systems including the central nervous system (CNS), immune system, and hypothalamic-pituitary-adrenal (HPA) axis [1,2,3]. These fundamental systems develop in a sexually dimorphic manner which is primarily due to differences in circulating gonadal hormones [4,5,6]. The sexually dimorphic nature of the maturation of these systems makes puberty a period of development that is particularly sensitive to stressors, influencing the pathogenesis of neurodevelopmental and neurodegenerative disorders (NDs) later in life, in a sex-specific manner [7,8]. Although the underlying mechanisms explaining the effects of pubertal stress exposure on these disorders remains unclear, the gut microbiome is a potential mechanism involved in their pathogenesis. The gut microbiome hosts trillions of microorganisms that influence the development and functioning of the CNS, immune system, and HPA axis [9,10,11]. Moreover, alterations to the gut microbiome have been associated with the development of neurodevelopmental and NDs [12,13,14]. However, the potential link between puberty, the gut microbiome, neurodegeneration, and neurodevelopmental disorders has not been extensively explored. As such, the purpose of this review is to summarize our current understanding of the gut microbiome, puberty, and neurodegeneration. We also examine how alterations to the gut microbiome during puberty may influence the pathogenesis of neurodevelopmental and NDs. Furthering our understanding of how puberty and the gut microbiome are associated with these disorders may allow for the development of therapeutic strategies that can prevent or mitigate their effects on brain and behavioral functioning.

## 2. Puberty

### 2.1. Timing and Sex Differences

Puberty is a critical period of development marked by sexual maturation, the development of secondary sexual characteristics, the activation of the hypothalamic–pituitary–gonadal axis, and the production of gonadal steroid hormones [3]. The hypothalamic–pituitary–gonadal axis initiates puberty by increasing the pulsatile gonadotropin releasing hormone (GnRH) secretion in the hypothalamus [15]. A key mechanism mediating the stimulation of GnRH neurons is the neuropeptide kisspeptin. Kisspeptin is a product of the *kiss-1* gene and has been shown to directly stimulate GnRH release by binding to GPR54, a G protein-coupled receptor that is located on GnRH neurons [16]. Metabolic (i.e., leptin, body fat), photoperiodic (i.e., melatonin), and environmental factors (i.e., stress) also contribute to the activation of GnRH neurons and the timing of pubertal onset [17,18,19]. The release of GnRH stimulates the production of luteinizing hormone and follicle stimulating hormone from the anterior pituitary into the bloodstream. Luteinizing hormone and follicle stimulating hormone then stimulate the gonads to initiate the maturation of gametes (i.e., gametogenesis) and the production of gonadal steroid hormones such as estradiol, progesterone, and androgens [20,21].

Increased activation of the hypothalamic–pituitary–gonadal axis and circulating gonadal steroid hormones play a central role in the development of secondary sexual characteristics (i.e., enlarged breasts and pubic hair in females and testicular enlargement and pubic hair in males) and the ability of an organism to reproduce (i.e., menarche in females and spermarche in males) [22,23,24,25]. In humans, this period of maturation typically begins around the ages of 8–13 in females and 9–14 in males [26]. The timing of puberty in animals can vary depending on the housing conditions and the strain of the animal. A non-invasive approach to identify pubertal onset in mice is vaginal opening in females and preputial descent (i.e., separation of the prepuce to the glans penis) in males [27,28]. It is estimated that CD1 and C57B1/6 female mice housed in single sex rooms demonstrate vaginal opening approximately 30 days following birth and have their first estrous cycle 20 days post vaginal-opening (Holder & Blaustein, 2014; Ismail and Blaustein, unpublished observations). Identifying pubertal onset through preputial separation in male mice is more difficult, however, measurements of scrotum width in six-week-old male CD1 mice indicate that the scrotum does not reach adult size until they are eight weeks old (Murray, Butcher, Kearns, Lamba, Stinzi & Ismail, in preparation).

### 2.2. Brain Reorganizing and Remodeling

Puberty is also a period during which the brain undergoes significant reorganizing and remodeling [3]. More specifically, the CNS undergoes significant functional and structural remodeling during this critical period of development including changes in both grey and white matter volumes [29]. Alterations in grey and white matter volumes vary by sex and are associated with the onset of gonadarche, suggesting that circulating gonadal steroid hormones play a role in brain development [30]. Grey matter volume (GMV) follows an inverted U-shape trajectory, with peak GMV being attained at the age of 11 for girls and 12 for boys in the frontal, temporal, and parietal lobes [30]. GMV increases in childhood and reaches its peak in adolescence due to dendritic arborization and synaptogenesis. GMV then steadily decreases into adulthood due to synaptic pruning [30].

Males typically have greater GMV then females while females have greater gray matter density then males, an effect that is primarily driven by differences in circulating gonadal steroid hormones [4]. Testosterone is associated with increases in global GMV, while estradiol is associated with decreases in global GMV [31]. The influence of circulating gonadal steroid hormones on GMV could also be region-specific. Testosterone has been associated with increases in GMV in the amygdala and decreases in GMV in the hippocampus, while estradiol has been associated with increases in limbic GMV [32]. Moreover, only males show increases in GMV in the amygdala during puberty while only females show increases in GMV in the hippocampus and striatum during puberty [32]. In contrast to GMV, white matter volume typically follows a linear trajectory from childhood to adolescence and then steadily stabilizes into adulthood [33]. Males have greater global white matter volume then females during puberty due to steeper age-related increases in axonal calibre [34,35]. Moreover, testosterone is associated with increases in white matter volume, while estradiol has been associated with either having no effect or a negative effect on white matter volume [36,37]. As such, circulating gonadal steroid hormones during puberty can have long-term effects on neurological changes and neural functioning into adulthood.

### 2.3. Maturation of the HPA Axis and Vulnerability to Stress

A critical neuroendocrine system that develops and matures during puberty is the HPA axis [2]. The HPA axis is the body’s primary stress processing system that has numerous adaptive physiological processes aimed at regulating allostatic load and maintaining homeostasis in response to a stressor [38]. These processes include redirecting energy resources, increasing vasoconstriction, cognition and metabolism, and regulating immune and reproductive functions [39,40]. When initially exposed to stressful stimuli, a rapid stress response mediated by the sympathetic nerves and the adrenal medulla is activated, inducing the release of catecholamines (i.e., epinephrine, norepinephrine) which results in increased heart rate and blood pressure along with decreased intestinal motility and bronchiolar dilatation [41,42]. This rapid stress response is followed by a slower stress response mediated by the HPA axis.

The HPA axis response begins with excitatory signals from the prefrontal cortex (PFC) and the amygdala to the paraventricular nucleus of the hypothalamus [43,44]. Stimulation of the paraventricular nucleus of the hypothalamus induces the release of corticotrophin-releasing hormone and arginine vasopressin, which bind to receptors in the anterior pituitary gland resulting in the release of adrenocorticotropic hormone (ACTH) [45]. ACTH then binds to receptors in the adrenal cortex, resulting in the synthesis of glucocorticoids (GC) (i.e., cortisol in humans and corticosterone in mice and rats) [45]. GC levels increase rapidly in the bloodstream following exposure to stress with peak levels being attained 20–40 min following initial stress exposure [46]. However, peak latency can vary depending on various factors such as sex, age, psychosocial factors, and previous exposure to a stressor [47,48,49,50]. GC have the ability to regulate their own production through a negative feedback mechanism. Mineralocorticoid (MRs) and glucocorticoid receptors (GRs) throughout the hypothalamus, pituitary, medial prefrontal cortex, and hippocampus play critical roles in this negative feedback mechanism [51]. MRs demonstrate a high affinity for GC and are activated once GC levels are low (i.e., basal levels) [52]. However, when GC levels increase, MRs become saturated resulting in the activation of GRs which have a lower affinity for GC. Once GRs are activated, signals are sent to the hypothalamus and pituitary to inhibit the production of GC [52]. This negative feedback mechanism is critical as it permits the rapid downregulation of GC synthesis, allowing the body to return to homeostasis following exposure to a stressor.

There are age and sex differences in HPA axis reactivity which are dependent on the strain and species of the animal being analyzed. For example, inbred prepubertal male mice exposed to acute restraint stress show greater HPA axis reactivity, as shown through either a more prolonged (i.e., C57BL/6) or greater (i.e., BALB/c) CORT response in comparison to their adult counterparts [5]. Inbred prepubertal female mice show similar HPA axis reactivity in comparison to their adult counterparts. Conversely, alterations to HPA axis reactivity in outbred mice (i.e., Swiss Webster) are only observed in adult female mice with these mice demonstrating a greater CORT response in comparison to their prepubertal counterparts [5]. Prepubertal male rats exposed to acute stressors (i.e., hypoxia, restraint, foot shock) show a more prolonged ACTH and CORT response in comparison to their adult counterparts [53,54,55,56]. When exposed to chronic restraint stress, prepubertal male rats demonstrate a greater stress response which is followed by a quicker return to baseline in comparison to their adult counterparts [55]. Adult male rats exposed to a homotypic stressor show a habituated stress response that is not observed in adolescent male rats [57,58,59]. Furthermore, adult female rats demonstrate a greater stress response in comparison to their adult male counterparts [60]. Taken together, these results demonstrate that HPA axis reactivity is highly dependent on the sex (i.e., male and female), species (i.e., mice and rats), and strain (i.e., inbred and outbred) of the subjects being analyzed.

Interestingly, age and sex differences in HPA axis reactivity emerge during puberty, suggesting a potential role of circulating gonadal steroid hormones (i.e., estradiol, testosterone, progesterone). Testosterone appears to decrease HPA axis reactivity while estradiol appears to increase it [61]. For example, gonadectomized female rats exposed to restraint stress display a decrease in ACTH and CORT concentrations in comparison to intact females, an effect that is reversed following estradiol treatment [62,63,64,65]. Conversely, gonadectomized male rats display an increase in ACTH and CORT concentrations following exposure to a stressor, an effect that is reversed following androgen treatment [66,67,68,69]. Thus, circulating gonadal steroid hormones play a critical role in HPA axis function and the increase in these hormones during puberty is essential for the development of adult-like patterns of HPA axis reactivity [70].

Maturation of the HPA axis during puberty is associated with a sex-dependent increase in vulnerability to stress-related disorders. Mental illnesses such as anxiety, depression, psychosis, eating disorders, bipolar disorder, substance abuse, and personality disorders, predominantly emerge during adolescence and puberty [71,72,73]. Moreover, the incidence of these mental illnesses are sex-dependent, with females showing a higher prevalence of eating disorders, anxiety and depression while males show a higher prevalence of psychosis and substance abuse [74,75,76,77,78]. Sex differences for many of these disorders also emerge during puberty with pubertal status (Tanner Stage III) being a better predictor of these sex differences then chronological age [79,80,81].

Factors contributing to the increased incidence of mental illness during puberty remain unclear, however, it is believed that exposure to stressors may interfere with the development of the HPA axis [82]. An atypical development of the HPA axis could reduce an individual’s ability to cope with stressors which may, in turn, increase vulnerability to stress-related disorders [38]. For example, repeated exposure to stressors during puberty can result in the sensitization of the HPA axis and the overproduction of GC [83,84]. The overproduction of GC could then damage key brain regions (i.e., hippocampus, amygdala, PFC) responsible for the regulation of the HPA axis [51,85,86]. Dysregulation of the HPA axis could then result in the chronic overproduction of GC which could increase susceptibility to mental illness associated with chronic stress (i.e., depression, anxiety, and substance abuse) [87,88,89]. Alternatively, it is possible that exposure to stressors during puberty can result in the blunting of the HPA axis [90]. Pubertal stress exposure may increase the expression of GRs which could facilitate the downregulation of GC and the blunting of the HPA axis [91]. Consequently, the blunting of the HPA axis could increase susceptibility to disorders such as post-traumatic stress disorder and personality disorders (i.e., antisocial and borderline personality disorder) [92,93,94].

### 2.4. Maturation of the Immune System and Enduring Effects of Lipopolysaccharide (LPS)

The immune system is a complex system of cells and proteins responsible for protecting an organism from viruses, bacteria, and other pathogens [1]. Like the HPA axis, the immune system also undergoes significant maturation during puberty [95]. The immune system of vertebrates is made up of two parts, the innate and adaptive immune systems [1]. The innate immune system is the body’s first line of defense, which has a non-specific response to pathogens that are evolutionarily conserved such as, bacterial, fungal, viral, or foreign proteins. Innate immune responses to a pathogen include inflammation, phagocytosis, and lysis [96,97] (Figure 1A). The adaptive immune system develops throughout the lifespan and utilizes B and T cells to recognize and remember foreign pathogens (Marshall et al., 2018). Adaptive immune responses destroy invading pathogens either indirectly through the secretion of antibodies by B cells (i.e., antibody response) or directly by T cells (i.e., cell mediated immune response) [98] (Figure 1B,C). An essential component of both the innate and adaptive immune systems is the production of cytokines. Cytokine is a general term used for a family of small proteins which include chemokines, interferons, interleukins, lymphokines, and tumor necrosis factors. Cytokines play critical roles in cell signaling and in the regulation of inflammation in response to invading pathogens [99].

A common method to activate the immune system in the laboratory is through the administration of lipopolysaccharide (LPS). LPS is a bacterial endotoxin located on the outer membranes of Gram-negative bacteria (i.e., *Escherichia coli*) and is a potent stimulator of the innate immune system [100]. In the periphery, LPS binds to Toll-like receptor 4 which is predominantly expressed on immune cells [101]. LPS can also influence the CNS by crossing the blood–brain barrier (BBB) and binding to Toll-like receptor 4 residing on microglia (i.e., primary innate immune cells of the brain) [100,102,103,104]. The stimulation of Toll-like receptor 4 induces a cascade of intracellular events that results in the stimulation of nuclear factor B (NF-κB) [100]. The stimulation of NF-κB then results in the synthesis of prostaglandins (i.e., prostaglandin E2), cyclooxygenase, nitric oxide, and pro- (i.e., TNFα, IL1β, IL12) and anti-inflammatory cytokines (i.e., IL10, IL4, IL9) [102,103,104]. Moreover, LPS administration in animals has been shown to induce anxiety, depression, neurodegeneration, and neurodevelopmental disorders [105,106,107]. Thus, LPS is an ideal candidate to examine the role that the immune system plays in various disorders.

LPS administration during puberty has enduring effects on brain functioning and behaviors. Previous research from our laboratory and others has shown that pubertal LPS treatment, in mice suppresses sexual receptivity in females, and increases anxiety-like behaviors in males and depression-like behaviors in females, in adulthood [107,108,109,110]. Moreover, pubertal LPS has been shown to decrease estrogen receptor-α and increase c-fos expression in adulthood [108,111]. Pubertal LPS treatment also has programming effects on immune and HPA axis reactivity. For example, pubertal LPS treatment permanently decreases GR expression in the paraventricular nucleus of the hypothalamus of adult male mice [112]. Furthermore, LPS treatment during puberty followed by a secondary immune challenge in adulthood results in an attenuated immune response as shown through decreases in peripheral IL6 and IFNγ concentrations and decreases in IL1β, TNFα, and IL6 mRNA expression in the PFC [113]. Pubertal LPS treatment also causes enduring learning and spatial memory deficits in both male and female mice and increases Parkinson-like behaviors in male, but not in female mice, indicating an increased susceptibility to neurodegeneration [109,111,114].

There are also age and sex differences in immune responsivity following LPS treatment, due in part to the immune-enhancing effects of estrogens and the immune-suppressing effects of androgens and progesterone [6,115,116]. In general, pubertal LPS treatment in mice induces a hypo-responsive immune response when compared to adult mice. For example, adult mice display greater peripheral pro-inflammatory cytokine concentrations compared to pubertal mice, while pubertal mice display greater peripheral anti-inflammatory cytokine concentrations compared to adult mice 10 h following LPS treatment [117]. However, pubertal male mice display greater IL1β, TNFα, and IL6 mRNA expression in the PFC compared to adult male mice 2 h following LPS treatment [104], and adult male mice display greater cytokine mRNA expression compared to pubertal male mice 8 h following LPS treatment [104]. These age and sex differences in cytokine expression are also associated with greater sickness behaviors and hypothermia in adult male mice compared to their pubertal counterparts [117]. Therefore, behavioral and physiological responses to an immune challenge are highly dependent on age and sex.

## 3. Neurodegeneration

NDs such as Parkinson’s disease (PD), Alzheimer’s disease (AD), multiple sclerosis (MS), amyotrophic lateral sclerosis (ALS), and Huntington’s disease are a common cause of morbidity and mortality in the elderly population [118]. Aging is the primary risk factor for NDs and with increased life expectancy worldwide, the prevalence of these disorders is increasing [119,120,121]. Neurodegeneration is commonly defined as the progressive loss of neuronal function in the CNS, resulting in impairments related to motor skills (e.g., gait, ataxia), cognition (e.g., memory, executive functions) and behaviors (e.g., disinhibition, apathy) [122,123,124]. The progressive loss of neuronal function in NDs is typically caused by abnormal protein aggregations (i.e., amyloidosis, tauopathies, synucleinopathies, and transactivation response DNA binding protein 43 proteinopathies), which can induce oxidative stress, excitotoxicity, mitochondrial dysfunction, and neuroinflammation, ultimately resulting in apoptosis [125,126,127,128,129,130].

The pathology of NDs is influenced by various factors. Abnormal protein aggregations are a hallmark of NDs with each ND being characterized by the aggregation of specific proteins. Examples of abnormal protein aggregations include amyloid beta (Aβ) and tau in AD, tau in Pick’s disease, alpha-synuclein in PD, and transactivation response DNA binding protein 43 (TDP-43) in ALS [128] (Figure 2). Although NDs differ in their clinical presentations and histopathological features, the aggregation of pathological proteins induces similar neurodegenerative processes. For example, Aβ, tau, alpha-synuclein, and TDP-43 are known to localize on the mitochondrial membrane and prevent neurons from functioning normally by causing mitochondrial damage, disrupting the election transport chain, increasing the production of reactive oxygen species and inducing persistent neuroinflammation and glutamate excitotoxicity [131,132,133,134,135,136]. Interestingly, the knockdown of these proteins does not cure or ameliorate neurodegeneration but rather results in severe motor (i.e., motor neuron degeneration), cognitive (i.e., spatial and long-term memory), and behavioral abnormalities (i.e., anxiety-like behavior) [137,138,139,140]. These deficits reflect the essential role these proteins play in regulating the morphology and physiology of neurons (i.e., neural growth and repair, cytoskeleton scaffolding, regulation of gene expression, and neurotransmitter release) [141,142,143,144]. While abnormal protein aggregations are fundamental to the pathology of NDs, other mechanisms may also be involved.

NDs differ in their symptoms due in part to differences in the cellular and neuroanatomical distribution of the proteins implicated in these disorders [128]. For example, the early stages of AD is typically characterized by cell loss and neurofibrillary tangles in the neocortex, hippocampus, entorhinal cortex, amygdala, and basal nucleus of Meynert [145]. Atrophy in these brain regions could result in memory loss, praxis, visuospatial impairments, and executive dysfunction; symptoms that are commonly observed in AD patients [146]. Conversely, the early stages of PD is typically characterized by the loss of dopamine producing cells in the basal ganglia (i.e., caudate nucleus, putamen, globus pallidus, subthalamic nucleus, and substantia nigra), which causes severe motor dysfunction (i.e., tremors, bradykinesia, muscular rigidity) in patients suffering from PD [147]. However, as NDs progress, a network of brain regions is affected, resulting in significant overlap between NDs in their clinical features. As such, it is not uncommon for NDs to have comorbidities with other NDs and psychiatric issues [128]. For example, AD and PD often have comorbidities with dementia with Lewy Bodies, progressive supranuclear palsy, vascular dementia, cerebral amyloid angiopathy, and depression [148,149].

### Sex Differences in Neurodegeneration

There are definite sex differences in the prevalence and clinical presentations of NDs. For example, AD is more prevalent in women (2:1) while PD more prevalent in men (2:1) [150,151,152]. Men suffering from AD tend to show more aggressive behaviors, have more comorbidities, and higher mortality rates than women, while women tend to show more affective symptoms (i.e., apathy, depression, irritability, anxiety), cognitive deterioration and higher survival rates than men [153,154]. In terms of PD, females typically show a slower rate of decline, fewer symptoms, and a delayed onset of PD, while males tend to display more aggressive motor dysfunction (i.e., postural instability, falling, gait disturbances) and impairments in executive function and reduced processing speed [155,156]. There are also sex differences in the prevalence and clinical presentations of other NDs such as ALS, MS, frontotemporal dementia, and Huntington’s disease [157,158,159]. Therefore, sex is a critical factor in the pathology of NDs and should be taken into account when examining the etiology of NDs.

It is well established that immune dysfunction plays a critical role in the etiology of NDs and sex differences related to the immune system may mediate sex differences observed in NDs. Of particular interest is sex differences in the number and morphology of microglia. Research in rats has shown that males have more microglia than females at postnatal day 4 (P4) in the parietal cortex, hippocampus, and amygdala [160]. However, this effect is reversed at adolescence (P30), with females displaying a greater number of activated microglia then males [160]. Sex differences in microglia number and morphology suggests that there may be sex-specific periods in development (i.e., puberty) where the over-activation of microglia can have enduring effects on microglial and neuronal function [157]. Moreover, males tend to have a higher incidence of NDs earlier in life (i.e., ALS, PD, schizophrenia) whereas females tend to have a higher incidence of NDs later in life (i.e., AD, schizophrenia), supporting the notion that perturbations of microglial function during critical periods of development may influence the development of NDs later in life [7,8,161,162]. Thus, higher number of activated microglia during critical periods of development could be harmful and may influence the development of NDs in males and females.

Sex differences in the transcriptome of microglia in adulthood could also explain sex differences observed in NDs [163,164]. Profiling of microglia in 3-, 12-, and 24-month old male and female mice revealed sex differences in the transcriptome of microglia at all time points with the greatest sex differences observed in 24-month old mice [165]. Moreover, single-nuclei RNA sequencing in the prefrontal cortex revealed large microglial transcriptomic differences between AD and control human brains [166]. Interestingly, a greater number of microglia associated with AD was observed in female brains while a greater number of microglia not associated with AD was observed in male brains [166]. These results suggest that the transcriptome of microglia becomes sexually divergent with age and that these transcriptomic sex differences could influence the development of NDs. Another factor that has been strongly linked to the development of NDs is the gut microbiome.

## 4. Gut Microbiome

The gut microbiome hosts trillions of microorganisms including bacteria, archaea, viruses and eukaryotic microbes. These microorganisms reside along the intestinal tract (i.e., esophagus, stomach, and intestine) and have various functions aimed at maintaining physiological homeostasis. Functions include vitamin and nutrient synthesis, carbohydrate fermentation, regulating immune function, and protecting against pathogens [167,168]. The understanding of the function and structure of the gut microbiome in health and disease has greatly increased over the years, due primarily to technological advancements (i.e., 16s RNA sequencing) in the analysis of microbial composition [169]. In healthy individuals, the gut microbiome predominately consists of bacterial species from the *Bacteroidetes* and *Firmicutes* phyla. There are also less abundant amounts of bacterial species from the *Proteobacteria*, *Actinobacteria*, *Verrucomicrobia*, and *Fusobacteria* phyla [170]. There are also varying amounts of bacteria depending on the region of the intestinal tract being examined. For example, the colon has a high density of bacteria from the *Bacteroidaceae*, *Prevotellaceae*, *Rikenellaceae*, *Lachnospiraceae* and *Ruminococcaceae* families, while the small intestine has a high density of *Lactobacillaceae* and *Enterobacteriaceae* families [171,172]. Various factors can influence the composition of the gut microbiome such as sex, genetics, antibiotics, probiotics, ethnicity, diet, and bacterial infections [173,174,175,176,177,178]. An imbalance in the gut microbial community (i.e., dysbiosis) can have harmful effects on brain functioning and behavior and is associated with various disorders such as autism, depression, anxiety, AD, and PD [179,180,181,182].

The bidirectional communication between microbiota and the brain is referred to as the ‘microbiota-gut-brain axis’ [183]. The microbiota-gut-brain axis is composed of multiple pathways including the CNS, autonomic nervous system, and enteric nervous system [184,185]. The autonomic system includes sympathetic and parasympathetic branches consisting of afferent and efferent fibers responsible for various involuntary physiological processes (i.e., heart rate, blood pressure, digestion) [186]. Afferent signals begin in the intestinal lumen and travel to the CNS through spinal, vagal, and enteric pathways, while efferent signals begin in the CNS and travel to the intestinal wall [184]. The gut microbiota can also influence the CNS through the production of several bioactive molecules such as cytokines, prostaglandins, and microbial antigens (i.e., LPS) [10]. These molecules can cross the BBB and directly influence the functioning of the CNS [187]. Thus, both humoral and neural pathways are involved in the microbiota-gut-brain axis and can have a profound influence on brain functioning and behavior.

### 4.1. Role of Microbiota in Neurodevelopment

The gut microbiota plays a vital role in neurodevelopmental processes such as neurogenesis, myelination, maturation of microglia, and BBB formation [9,11,188,189]. For example, germ free (i.e., mice raised without microbiota) and antibiotic-treated mice display reduced expression of the tight junction proteins occludin and claudin-5 in the hippocampus, suggesting increased BBB permeability [190,191]. Furthermore, antibiotic treatment in mice results in a decrease in bromodeoxyuridine in the hippocampus, indicating a reduction in hippocampal neurogenesis [192]. Similarly, the absence or alteration of the gut microbiota can result in abnormal myelination and microglial function. For instance, germ free mice display a high number of immature microglia in several brain regions (i.e., cortex, corpus collosum, hippocampus, cerebellum), an effect that is replicated in antibiotic-treated mice [9]. Moreover, germ free mice display abnormal myelination of axons within the PFC [188,193]. Normal functioning of these neurodevelopmental processes can be restored through the administration of microbial metabolites (short chain fatty amino acids; SCFA) and probiotics [9,190,192,194]. Therefore, the gut microbiota plays a significant role in the regulation of neurodevelopment and can either have protective or harmful effects on the CNS during development.

Alterations to the gut microbiome is associated with neurodevelopmental disorders such as autism spectrum disorder and schizophrenia [12,13]. Gut dysbiosis has been reported in autism spectrum disorder patients with these patients having elevated levels of *Proteobacteria*, *Lactobacillus*, *Bacteroides*, *Desulfovibrio*, and *Clostridium* along with decreased levels of *Bifidobacterium*, *Blautia*, *Dialister*, *Prevotella*, *Veillonella* and *Turicibacter*, in comparison to controls [195]. Similarly, schizophrenic patients have elevated levels of *Succinivibrio*, *Megasphaera*, *Collinsella*, *Clostridium*, *Klebsiella*, and *Methanobrevibacter* along with decreased levels of *Coprococcus*, *Roseburia*, and *Blautia,* in comparison to controls [196]. Moreover, these neurodevelopmental disorders are often accompanied with gastrointestinal disorders including constipation, diarrhea, abdominal pain, celiac disease, and irritable bowel syndrome [197,198,199]. Providing treatments that target restoring gut microbiota homeostasis (i.e., probiotics) has also been shown to ameliorate the symptoms of ASD and schizophrenia [200,201,202,203]. Taken together, these findings demonstrate the impact that the microbiota can have on brain functioning, and possibly mediate the onset and progression of neurodevelopmental disorders. However, the influence of the gut microbiota on neurodevelopment during puberty and adolescence remains largely uninvestigated.

### 4.2. Microbiota and Stress

Growing evidence suggests that the gut microbiome can influence the maturation and reactivity of the HPA axis [10]. Germ free mice exposed to 1 h of restraint stress display greater levels of plasma ACTH and CORT in comparison to specific pathogen free mice [204]. Moreover, reconstitution with commensal bacteria at 3 weeks of age reverses the elevated HPA axis response observed in germ free mice. However, reconstitution at a later stage (i.e., before 6 weeks of age) has no effect on the HPA axis response [204]. The effects of the gut microbiome on HPA axis reactivity are not limited to blood markers. Germ free mice also display elevated levels of several glucocorticoid receptor pathway genes (i.e., Slc22a5, Aqp1, Stat5a, Ampd3, Plekhf1, and Cyb561) in the hippocampus [205]. Although the mechanisms underlying the effects of microbiota on HPA axis responsiveness are not fully understood, it is believed that gut dysbiosis (i.e., induced by exposure to stressors) increases the production of bioactive molecules that can directly influence the HPA axis [206]. For instance, gut dysbiosis upregulates cytokines (i.e., TNFα, IL1β, IL6) and prostaglandins (i.e., prostaglandin E2) which can subsequently cross the BBB and activate the HPA axis [207,208,209,210]. Moreover, both LPS and peptidoglycan (i.e., components of the cell wall of Gram-negative bacteria) are upregulated in response to gut dysbiosis and have been shown to be potent activators of the HPA axis [211,212,213]. As such, the gut microbiota can influence HPA axis reactivity and play a crucial role in the programming of the HPA axis.

Multiple lines of research have also demonstrated an association between microbiota and stress-related disorders such as anxiety and depression. Both anxiety and depression have high comorbidities with gastrointestinal disorders such as irritable bowel syndrome, Crohn’s disease, celiac disease, and ulcerative colitis [214,215,216]. Moreover, research with germ free mice has shown that the absence of microbiota results in a decrease in anxiety-like behaviors while germ free rats display an increase in anxiety-like behaviors [217,218]. Interestingly, colonizing germ-free Swiss Webster mice with microbiota from Balb/C mice increases anxiety-like behaviors, while colonizing germ free Balb/C mice with microbiota from Swiss Webster mice decreases anxiety-like behaviors [219]. Similar findings are observed when examining the effects of gut microbiota on depression-like behaviors. For example, gut dysbiosis induced by chronic unpredictable mild stress in mice increases depression-like behaviors. Moreover, transferring microbiota from stressed mice to naïve mice results in an increase in depression-like behaviors in the naïve mice [220].

### 4.3. Microbiota and Neurodegeneration

Gut dysbiosis has been implicated in various neurodegenerative processes including the production of amyloid proteins, inflammation, oxidative stress, impaired SCFA synthesis, and increased intestinal and BBB permeability [14,190,221,222,223] (Figure 3). Recent research has shown that AD patients suffer from dysbiosis as demonstrated by increased levels of *Ruminococcaceae*, *Enterococcaceae*, and *Lactobacillaceae* along with decreased levels of *Bacteroidaceae*, *Veillonellaceae*, and *Lachnospiraceae*, in comparison to controls [224]. Furthermore, antibiotic-induced dysbiosis in a mouse model of AD has been shown to reduce neuroinflammation and amyloidosis, implicating microbiota in the pathogenesis of AD [225]. Similarly, PD patients have been reported to suffer from dysbiosis with increased levels of *Lactobacillus*, *Akkermansia*, and *Bifidobacterium* and decreased levels of *Lachnospiraceae* and *Faecalibacterium*, in comparison to controls [226]. Moreover, in a rotenone-induced mouse model of PD, wild-type mice display greater motor deficits (i.e., motor strength and coordination) in comparison to germ free mice, further supporting the role of microbiota in the development of PD [227].

It is theorized that proteins involved in NDs, such as Aβ and alpha-synuclein, can aggregate and spread throughout the brain in a prion-like manner (i.e., neuron-to-neuron propagation) [228,229]. However, there is continuous debate regarding the site of origin for the pathological aggregation of these proteins. Both Aβ and alpha-synuclein aggregations have been observed in the gut prior to spreading to the CNS, suggesting that both AD and PD pathology may originate in the gut [230,231]. Aggregates of alpha-synuclein can be found within enteroendocrine cells (i.e., sensory cells of the gut) which synapse with enteric nerves. From enteric nerves, alpha-synuclein aggregates can enter the vagus nerve allowing for transportation to the brain [232]. Similarly, Aβ aggregates have been observed in the vagus nerve, suggesting that a similar pathway may be used for the spreading of Aβ to the CNS [233]. Research in mice has also shown that vagotomy reduces the pathologic spreading of Aβ and alpha-synuclein in the CNS, further supporting the role of the vagus nerve as a potential pathway involved in NDs [234,235]. As such, the gut microbiome may not only influence the pathogenesis of NDs, but could also be the site of origin for NDs.

## 5. Conclusions

Puberty is marked by the maturation of fundamental systems such as the HPA axis and the immune system along with significant reorganizing and remodeling of the CNS. These pubertal changes render the CNS particularly sensitive to stressors and increase its susceptibility to neurodevelopmental and neurodegenerative disorders later in life. Alterations to the gut microbiome have also been associated with neurodevelopmental and neurodegenerative disorders. However, the underlying mechanisms mediating the enduring effects of microbiota on neurodegenerative and neurodevelopmental disorders remain unclear. Future research should investigate the mechanisms through which the gut and brain communicate along with determining the link between microbiota, neurodevelopment and neurodegenerative disorders. Moreover, a greater understanding of the effects of microbiota, during critical periods of development (i.e., puberty), in increasing the vulnerability to neurodevelopmental and neurodegenerative disorders is needed. Lastly, further research is required to determine whether treatments that target the microbiome (i.e., probiotics, prebiotics) can effectively prevent the development of neurodevelopmental and neurodegenerative disorders.

## Figures and Tables

**Figure 1 microorganisms-10-02163-f001:**
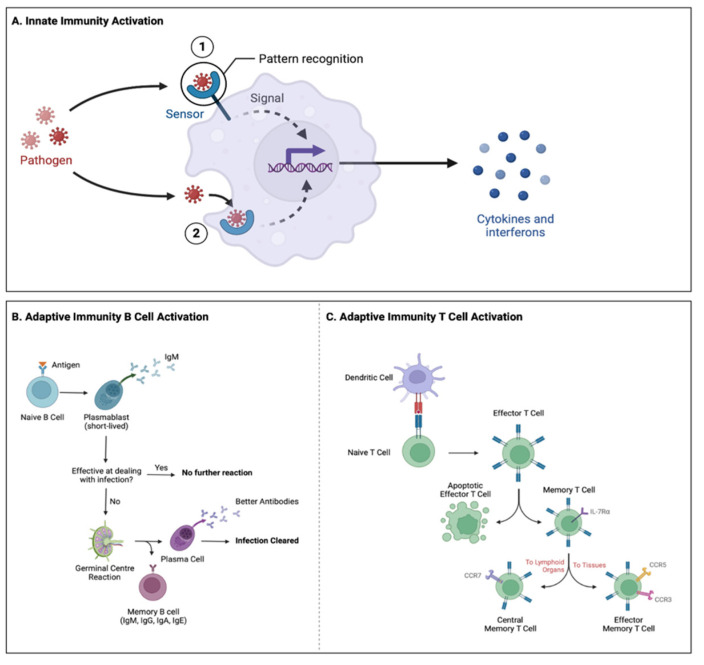
Innate and Adaptive Immunity Activation Process. (**A**) The innate immune system is the body’s first line of defense against foreign pathogens and consists of various types of cells such as monocytes, macrophages, neutrophils, mast cells, eosinophils, basophils, dendritic cells, and natural killer cells. The innate immune system is activated once pathogen-associated molecular patterns (PAMPs; i.e., LPS) or damage-associated molecular patterns (DAMPs; i.e., S100 proteins) are recognized by pattern recognition receptors (PRRs). PRRs have the ability to detect both (1) extracellular and (2) intracellular pathogens based on their cellular location. Once PRRs detect a pathogen they rapidly activate immune cells to produce cytokines and interferons to clear pathogens. The innate immune response can also clear pathogens through other processes such as destruction via phagocytosis, by natural killer cells and/or through the activation of the complement system. (**B**) The body’s second line of defense involves the adaptive immune system. Once exposed to a novel antigen, the antigen will bind to a B cell receptor (BCR) generated by V(D)J recombination which will result in the activation of the naïve B cell. Once activated, the naïve B cell will differentiate into antibody-secreting cells called plasmablasts. Immunoglobulin M (IgM) is the first antibody produced by plasmablasts to fight off the initial infection. If IgM is ineffective in fighting a pathogen, additional B cells will be generated through the germinal center as plasma cells and memory B cells. The B cells have optimized BCRs better equipped to fight off pathogens. Plasma cells can secrete antibodies for weeks following their activation and then migrate to the bone marrow where they can reside for long periods of time. Memory B cells circulate throughout the body on the lookout for antigens that bind to their BCR and quickly respond to remove the antigen if encountered. With every subsequent pathogen encounter, this cycle will repeat to further optimize the BCRs. (**C**) As part of the adaptive immune response, T cells can become activated through dendritic cells and other antigen presenting cells. Dendritic cells contain novel antigens from peripheral tissue which are presented to T cells for activation of the cell. Once activated, naïve T cells differentiate into effector T cells which can either directly induce apoptosis in an infected cell (i.e., cytotoxic T cell), activate other immune cells (i.e., helper T cell), or suppress an immune response (i.e., regulatory T cell). Once antigens have been cleared, effector T cells can further differentiate into memory T cells which are antigen-specific T cells. Memory T cells can then differentiate to central memory T cells and effector memory T cells. Effector memory T cells primarily localize in peripheral tissue and provide an immediate defense against reintroduced antigens while central memory T cells reside in secondary lymphoid tissues and sustain the immune response against the specific antigen (**A**) adapted from “Innate Immunity”, by Biorender.com (accessed on 16 October 2022). Retrieved from https://app.biorender.com/biorender-templates (accessed on 16 October 2022), (**B**,**C**) created with Biorender.com (accessed on 16 October 2022).

**Figure 2 microorganisms-10-02163-f002:**
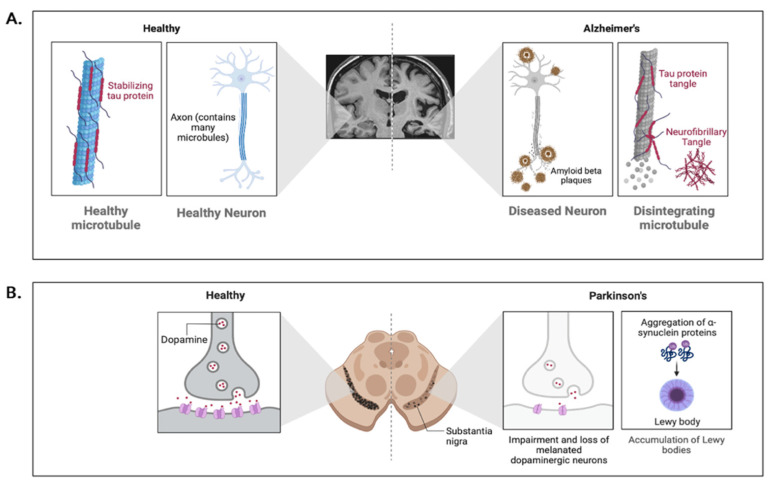
Abnormal Protein Aggregations in Alzheimer’s and Parkinson’s Disease. (**A**) Healthy neurons are supported internally by microtubules which are stabilized by the protein tau. In Alzheimer’s disease, abnormal accumulations of amyloid-beta are present which form plaques around neurons. The increase in amyloid-beta plaques causes an upregulation of abnormal conformations of tau which detach from microtubules and bind to other tau molecules forming neurofibrillary tangles. The increase in amyloid-beta plaques and neurofibrillary tangles severely impairs cell functioning and causes apoptosis. (**B**) Parkinson’s disease is characterized by the loss of dopaminergic neurons in the substantia nigra pars compacta. The loss of dopaminergic neurons is caused by Lewy body inclusions which consist of abnormal aggregations of alpha-synuclein. Lewy bodies drastically decrease dopamine synthesis by decreasing the synthesis and activity of tyrosine hydroxylase, a precursor to dopamine (**A**) adapted from “Alzheimer’s disease 2”, by Biorender.com (accessed on 16 October 2022). Retrieved from https://app.biorender.com/biorender-templates (accessed on 16 October 2022); (**B**) adapted from “Progression of Parkinson’s disease in substantia nigra”, by Biorender.com (accessed on 16 October 2022). Retrieved from https://app.biorender.com/biorender-templates (accessed on 16 October 22).

**Figure 3 microorganisms-10-02163-f003:**
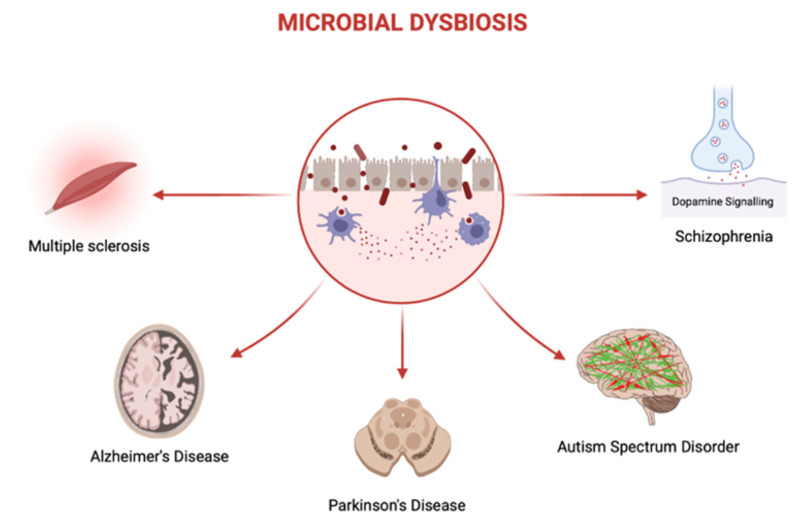
Neurodevelopmental and Neurodegenerative Disorders Linked with Dysbiosis. Microbial dysbiosis is defined as an imbalance of microbiota where there is a relative increase in pathogenic bacteria compared to beneficial ones. Microbial dysbiosis has been linked to several neurodevelopmental and neurodegenerative disorders including multiple sclerosis, Alzheimer’s disease, Parkinson’s disease, autism spectrum disorder, and schizophrenia. However, the underlying mechanisms explaining the effects of microbial dysbiosis on these disorders remain unclear (Adapted from “Dysbiosis”, by Biorender.com (accessed on 16 October 2022). Retrieved from https://app.biorender.com/biorender-templates (accessed on 16 October 2022).

## Data Availability

Not applicable.

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
