# Peer review of "Linking Puberty and the Gut Microbiome to the Pathogenesis of Neurodegenerative Disorders"

_microorganisms, 2022, doi:10.3390/microorganisms10112163_

Round 1

Reviewer 1 Report

The manuscript (review) summarizes recent knowledge about puberty, neurodegeneration and gut microbiome. The authors try to present a linkage between these problematics. The topic is interesting because there is still a need to explain the mechanism of action or association between gut and brain axis. The topic fits with the journal, but the manuscript fails in the summarized and presented information.

The introduction is missing. Why was this topic selected? What new should the review bring?

The absence of figures/schemes make it hard for the reader to follow the author´s ideas.

There is duplicity of the information in the text.

Some abbreviations are not explained in the text.

In the part dedicated to neurodegeneration some important papers by prof. Michal Novak (association between tau protein and Alzheimer´s disease) are missing.

Each section bring separate information. No linkage between the puberty, gut microbiome and neurodegeneration is presented in detail.

The information presented in each section is very general and basic. I recommend to modify each section. The authors should present associations and scientific studies dedicated to the mentioned topic/topics.

The manuscript can´t be published in the Journal Microorganisms.  

Author Response

Reviewer 1

General Comments

  • The introduction is missing. Why was this topic selected? What new should the review bring?

We agree and thank the reviewer for this comment. We have now included information about why we chose this topic and what new information this manuscript can provide in Introduction section 1.

“1. Introduction

Puberty is a period of development that is accompanied by the maturation of various fundamental systems including the central nervous system (CNS), immune system, and hypothalamic-pituitary-adrenal (HPA) axis (Brenhouse & Schwarz, 2016; Romeo, 2010; Sisk & Foster, 2004). These fundamental systems develop in a sexually dimorphic manner which is primarily due to differences in circulating gonadal hormones (Gennatas et al., 2017; Romeo, 2013; Taneja, 2018). The sexually dimorphic nature of the maturation of these systems makes puberty a period of development that is particularly sensitive to stressors, influencing the pathogenesis of neurodevelopmental and neurodegenerative disorders (NDs) later in life, in a sex-specific manner (Pinares-Garcia et al., 2018; Yanguas-Casás, 2017). Although the underlying mechanisms explaining the effects of pubertal stress exposure on these disorders remains unclear, the gut microbiome is a potential mechanism involved in their pathogenesis. The gut microbiome hosts trillions of microorganisms that influence the development and functioning of the CNS, immune system, and HPA axis (Cerdó et al., 2020; Erny et al., 2015; Farzi et al., 2018). Moreover, alterations to the gut microbiome have been associated with the development of neurodevelopmental and NDs (Mehra et al., 2022; Munawar et al., 2021; Roy Sarkar & Banerjee, 2019). However, the potential link between puberty, the gut microbiome, neurodegeneration, and neurodevelopmental disorders has not been extensively explored. As such, the purpose of this review is to summarize our current understanding of the gut microbiome, puberty, and neurodegeneration. We also examine how alterations to the gut microbiome, during puberty, may influence the pathogenesis of neurodevelopmental and NDs. Furthering our understanding of how puberty and the gut microbiome are associated with these disorders may allow for the development of therapeutic strategies that can prevent or mitigate their effects on brain and behavioral functioning.” (Page 1-2)

  • The absence of figures/schemes make it hard for the reader to follow the author´s ideas.

We thank the reviewer for this comment. We have now included figures to better illustrate the concepts being discussed. Figure 1 has been added in section 1.4. Maturation of the immune system and enduring effects of lipopolysaccharide (LPS), Figure 2 has been added in section 2. Neurodegeneration, and Figure 3 has been added in section 3.3. Microbiota and neurodegeneration.

Figure 1. Innate and Adaptive Immunity Activation Process

(A)       The innate immune system is the body’s first line of defence against foreign pathogens and consists of various types of cells such as monocytes, macrophages, neutrophils, mast cells, eosinophils, basophils, dendritic cells, and natural killer cells. The innate immune system is activated once pathogen-associated molecular patterns (PAMPs; i.e., LPS) or damage-associated molecular patterns (DAMPs; i.e. S100 proteins) are recognized by pattern recognition receptors (PRRs). PRRs have the ability to detect both (1) extracellular and (2) intracellular pathogens based on their cellular location. Once PRRs detect a pathogen they rapidly activate immune cells to produce cytokines and interferons to clear pathogens. The innate immune response can also clear pathogens through other processes such as destruction via phagocytosis, by natural killer cells and/or through the activation of the complement system (B) The body’s second line of defense involves the adaptive immune system. Once exposed to a novel antigen, the antigen will bind to a B cell receptor (BCR) generated by V(D)J recombination which will result in the activation of the naïve B cell. Once activated, the naïve B cell will differentiate into antibody-secreting cells called plasmablasts. Immunoglobulin M (IgM) is the first antibody produced by plasmablasts to fight off the initial infection. If IgM is ineffective in fighting a pathogen, additional B cells will be generated through the germinal center as plasma cells and memory B cells. The B cells have optimized BCRs better equipped to fight off pathogens. Plasma cells can secrete antibodies for weeks following their activation and then migrate to the bone marrow where they can reside for long periods of time. Memory B cells circulate throughout the body on the lookout for antigens that bind to their BCR and quickly respond to remove the antigen if encountered. With every subsequent pathogen en-counter, this cycle will repeat to further optimize the BCRs. (C) As part of the adaptive immune response, T cells can become activated through dendritic cells and other antigen presenting cells. Dendritic cells contain novel antigens from peripheral tissue which are presented to T cells for activation of the cell. Once activated, naïve T cells differentiate into effector T cells which can either directly induce apoptosis in an infected cell (i.e., cytotoxic T cell), activate other immune cells (i.e., helper T cell), or supress an immune response (i.e., regulatory T cell). Once antigens have been cleared, effector T cells can further differentiate into memory T cells which are antigen-specific T cells. Memory T cells can then differentiate to central memory T cells and effector memory T cells. Effector memory T cells primarily localize in peripheral tissue and provide an immediate defense against reintroduced antigens while central memory T cells reside in secondary lymphoid tissues and sustain the immune response against the specific antigen (Figure A adapted from “Innate Immunity”, by Biorender.com (2022). Retrieved from https://app.biorender.com/biorender-templates; Figures B and C created with Bioren-der.com).” (Page 6)

Figure 2. Abnormal Protein Aggregations in Alzheimer’s and Parkinson’s Disease

(A)       Healthy neurons are supported internally by microtubules which are stabilized by the protein tau. In Alzheimer’s disease, abnormal accumulations of amyloid-beta are present which form plaques around neurons. The increase of amyloid-beta plaques causes an upregulation of abnormal conformations of tau which detach from microtubules and bind to other tau molecules forming neurofibrillary tangles. The increase of amyloid-beta plaques and neurofibrillary tangles severely impairs cell functioning and causes apoptosis. (B) Parkinson’s disease is characterized by the loss of dopaminergic neurons in the substantia nigra pars compacta. The loss of dopaminergic neurons is caused by Lewy body inclusions which consist of abnormal aggregations of alpha-synuclein. Lewy bodies drastically decrease dopamine synthesis by decreasing the synthesis and activity of tyrosine hydroxylase, a precursor to dopamine (Figure A adapted from “Alzheimer’s disease 2”, by Biorender.com (2022). Retrieved from https://app.biorender.com/biorender-templates; Figure B adapted from “Progression of Parkinson’s disease in substantia nigra”, by Biorender.com (2022). Retrieved from https://app.biorender.com/biorender-templates).” (Page 9)

Figure 3. Neurodevelopmental and Neurodegenerative Disorders Linked with Dysbiosis

Microbial dysbiosis is defined as an imbalance of microbiota where there is a relative increase in pathogenic bacteria compared to beneficial ones. Microbial dysbiosis has been linked to several neurodevelopmental and neurodegenerative disorders including multiple sclerosis, Alzheimer’s disease, Parkinson’s disease, autism spectrum disorder, and schizophrenia. However, the underlying mechanisms explaining the effects of microbial dysbiosis on these disorders remains unclear (Adapted from “Dysbiosis”, by Biorender.com (2022). Retrieved from https://app.biorender.com/biorender-templates).” (Page 13)

  • There is duplicity of the information in the text.

We thank the reviewer for this comment. We have revised the manuscript to ensure that information is not repeated.

  • Some abbreviations are not explained in the text.

We are grateful for the reviewer’s comment. We have revised the manuscript and explained all abbreviations.

  • In the part dedicated to neurodegeneration some important papers by prof. Michal Novak (association between tau protein and Alzheimer´s disease) are missing.

We thank the reviewer for this comment. We have included references from Prof. Michal Novak in Neurodegeneration section 3.

“The pathology of NDs is influenced by various factors. Abnormal protein aggregations are a hallmark of NDs with each ND being characterized by the aggregation of specific proteins. Examples of abnormal protein aggregations include amyloid beta (Aβ) and tau in AD, tau in Pick’s disease, alpha-synuclein in PD, and transactivation response DNA binding protein 43 (TDP-43) in ALS (Dugger & Dickson, 2017; Harrington et al., 1991; Novak, 1994).” (Page 8)

  • Each section bring separate information. No linkage between the puberty, gut microbiome and neurodegeneration is presented in detail.

We thank the reviewer for this comment. We have revised the manuscript and made the link between puberty, gut microbiome, and neurodegeneration clearer.

  • The information presented in each section is very general and basic. I recommend to modify each section. The authors should present associations and scientific studies dedicated to the mentioned topic/topics.

We thank the reviewer for this comment. We have made sure that the manuscript includes scientific studies that are related to the topics being discussed.

Reviewer 2 Report

This paper provides a review to link puberty, gut microbiome, and neurodegenerative disorders. This is an important and interesting topic. This review details the mechanism to explain the association between them. While this paper is publishable, many of the references are more than 10 years old. When I read the paper, I doubt whether these contents are updated or not.  

Specific comments

1.      Page 3, lines 8 and 9.  “This rapid stress response is followed by a slower stress response mediated by the HPA axis.”.  It is not clear.

2.      Page 3. “GC have the ability to regulate their own production through a negative feedback mechanism.” Explain more.

3.      This paper provides a good review on linking puberty, the gut microbiome, and neurodegenerative disorders. However, many of the references are more than 10 years old. For example, the references in “Sex differences for many of these disorders also emerge during puberty with pubertal status (Tanner Stage III) being a better predictor of these sex differences then chronological age (Angold et al., 2003; Hayward & Sanborn, 2002; Patton et al., 1996)” are a long time ago. Can it be applied to the present?   The authors may consider removing some old references and citing more recent articles.

4.      Section 3.1. Replace “blood-brain barrier” with BBB.

Author Response

Reviewer 2 

Specific Comments

  • Page 3, lines 8 and 9. “This rapid stress response is followed by a slower stress response mediated by the HPA axis.”.  It is not clear.

We agree and thank the reviewer for this comment. We have made this information clearer in section 2.3. Maturation of the HPA axis and vulnerability to stress.

“GC have the ability to regulate their own production through a negative feedback mechanism. Mineralocorticoid (MRs) and glucocorticoid receptors (GRs) throughout the hypothalamus, pituitary, medial prefrontal cortex, and hippocampus play critical roles in this negative feedback mechanism (McCormick & Mathews, 2010). MRs demonstrate a high affinity for GC and are activated once GC levels are low (i.e., basal levels) (Herman et al., 2012). However, when GC levels increase, MRs become saturated resulting in the activation of GRs which have a lower affinity for GC. Once GRs are activated, signals are sent to the hypothalamus and pituitary to inhibit the production of GC (Herman et al., 2012). This negative feedback mechanism is critical as it permits the rapid downregulation of GC synthesis, allowing the body to return to homeostasis following exposure to a stressor.” (Page 3-4)

  • Page 3. “GC have the ability to regulate their own production through a negative feedback mechanism.” Explain more.

We agree and thank the reviewer for this comment. We included additional information in section 2.3. Maturation of the HPA axis and vulnerability to stress.

“When initially exposed to stressful stimuli, a rapid stress response mediated by sym-pathetic nerves and the adrenal medulla is activated, inducing the release of catechol-amines (i.e., epinephrine, norepinephrine) which results in increased heart rate and blood pressure along with decreased intestinal motility and bronchiolar dilatation (Chu et al., 2022; Goldstein, 2010). This rapid stress response is followed by a slower stress response mediated by the HPA axis.” (Page 3)

  • This paper provides a good review on linking puberty, the gut microbiome, and neurodegenerative disorders. However, many of the references are more than 10 years old. For example, the references in “Sex differences for many of these disorders also emerge during puberty with pubertal status (Tanner Stage III) being a better predictor of these sex differences then chronological age (Angold et al., 2003; Hayward & Sanborn, 2002; Patton et al., 1996)” are a long time ago. Can it be applied to the present? The authors may consider removing some old references and citing more recent articles.

We thank the reviewer for this comment. There are cases where older research articles are referenced because they are seminal papers that presently contribute to our understanding of the discussed topics. However, we have revised the manuscript and have included more recent references wherever possible.

  • Section 3.1. Replace “blood-brain barrier” with BBB.

We thank the reviewer for bringing this to our attention. We have resolved this issue in section 4.1. Role of microbiota in neurodevelopment.

“The gut microbiota plays a vital role in neurodevelopmental processes such as neurogenesis, myelination, maturation of microglia, and BBB formation (Cerdó et al., 2020; Erny et al., 2015; Hoban et al., 2016; Parker et al., 2020).” (Page 11)

Reviewer 3

General Comments

  • A table or a figure would help the reader to better appreciate the content of the manuscript.

We thank the reviewer for this comment. We have now included figures to better illustrate the concepts being discussed. Figure 1 has been added in section 1.4. Maturation of the immune system and enduring effects of lipopolysaccharide (LPS), Figure 2 has been added in section 2. Neurodegeneration, and Figure 3 has been added in section 3.3. Microbiota and neurodegeneration.

Figure 1. Innate and Adaptive Immunity Activation Process

(A)       The innate immune system is the body’s first line of defence against foreign pathogens and consists of various types of cells such as monocytes, macrophages, neutrophils, mast cells, eosinophils, basophils, dendritic cells, and natural killer cells. The innate immune system is activated once pathogen-associated molecular patterns (PAMPs; i.e., LPS) or damage-associated molecular patterns (DAMPs; i.e. S100 proteins) are recognized by pattern recognition receptors (PRRs). PRRs have the ability to detect both (1) extracellular and (2) intracellular pathogens based on their cellular location. Once PRRs detect a pathogen they rapidly activate immune cells to produce cytokines and interferons to clear pathogens. The innate immune response can also clear pathogens through other processes such as destruction via phagocytosis, by natural killer cells and/or through the activation of the complement system (B) The body’s second line of defense involves the adaptive immune system. Once exposed to a novel antigen, the antigen will bind to a B cell receptor (BCR) generated by V(D)J recombination which will result in the activation of the naïve B cell. Once activated, the naïve B cell will differentiate into antibody-secreting cells called plasmablasts. Immunoglobulin M (IgM) is the first antibody produced by plasmablasts to fight off the initial infection. If IgM is ineffective in fighting a pathogen, additional B cells will be generated through the germinal center as plasma cells and memory B cells. The B cells have optimized BCRs better equipped to fight off pathogens. Plasma cells can secrete antibodies for weeks following their activation and then migrate to the bone marrow where they can reside for long periods of time. Memory B cells circulate throughout the body on the lookout for antigens that bind to their BCR and quickly respond to remove the antigen if encountered. With every subsequent pathogen en-counter, this cycle will repeat to further optimize the BCRs. (C) As part of the adaptive immune response, T cells can become activated through dendritic cells and other antigen presenting cells. Dendritic cells contain novel antigens from peripheral tissue which are presented to T cells for activation of the cell. Once activated, naïve T cells differentiate into effector T cells which can either directly induce apoptosis in an infected cell (i.e., cytotoxic T cell), activate other immune cells (i.e., helper T cell), or supress an immune response (i.e., regulatory T cell). Once antigens have been cleared, effector T cells can further differentiate into memory T cells which are antigen-specific T cells. Memory T cells can then differentiate to central memory T cells and effector memory T cells. Effector memory T cells primarily localize in peripheral tissue and provide an immediate defense against reintroduced antigens while central memory T cells reside in secondary lymphoid tissues and sustain the immune response against the specific antigen (Figure A adapted from “Innate Immunity”, by Biorender.com (2022). Retrieved from https://app.biorender.com/biorender-templates; Figures B and C created with Bioren-der.com).” (Page 6)

Figure 2. Abnormal Protein Aggregations in Alzheimer’s and Parkinson’s Disease

(A)       Healthy neurons are supported internally by microtubules which are stabilized by the protein tau. In Alzheimer’s disease, abnormal accumulations of amyloid-beta are present which form plaques around neurons. The increase of amyloid-beta plaques causes an upregulation of abnormal conformations of tau which detach from microtubules and bind to other tau molecules forming neurofibrillary tangles. The increase of amyloid-beta plaques and neurofibrillary tangles severely impairs cell functioning and causes apoptosis. (B) Parkinson’s disease is characterized by the loss of dopaminergic neurons in the substantia nigra pars compacta. The loss of dopaminergic neurons is caused by Lewy body inclusions which consist of abnormal aggregations of alpha-synuclein. Lewy bodies drastically decrease dopamine synthesis by decreasing the synthesis and activity of tyrosine hydroxylase, a precursor to dopamine (Figure A adapted from “Alzheimer’s disease 2”, by Biorender.com (2022). Retrieved from https://app.biorender.com/biorender-templates; Figure B adapted from “Progression of Parkinson’s disease in substantia nigra”, by Biorender.com (2022). Retrieved from https://app.biorender.com/biorender-templates).” (Page 9)

Figure 3. Neurodevelopmental and Neurodegenerative Disorders Linked with Dysbiosis

Microbial dysbiosis is defined as an imbalance of microbiota where there is a relative increase in pathogenic bacteria compared to beneficial ones. Microbial dysbiosis has been linked to several neurodevelopmental and neurodegenerative disorders including multiple sclerosis, Alzheimer’s disease, Parkinson’s disease, autism spectrum disorder, and schizophrenia. However, the underlying mechanisms explaining the effects of microbial dysbiosis on these disorders remains unclear (Adapted from “Dysbiosis”, by Biorender.com (2022). Retrieved from https://app.biorender.com/biorender-templates).” (Page 13)

Specific Comments

  • Can you avoid the repetition of these similar words? Again. in the next sentences.

We agree and thank the reviewer for this comment. We have revised the manuscript and avoided the use of similar words wherever possible in the Abstract

“The purpose of this review is to summarize the current knowledge about puberty, neurodegeneration, and the gut microbiome. We also examine the consequences of pubertal exposure to stress and gut dysbiosis on the development of neurodevelopmental and neurodegenerative disorders. Understanding how alterations to the gut microbiome, particularly during critical periods of development (i.e., puberty), influences the pathogenesis of these disorders may allow for the development of therapeutic strategies to prevent them.” (Page 1)

  • Classically, the "gut-brain axis" refers to the two-way signaling between enteric nervous system and central nervous system. The more recent involvement of microbiota led to the concept of "microbiota-gut-brain axis". See: Margolis KG, Cryan JF, Mayer EA. The Microbiota-Gut-Brain Axis: From Motility to Mood. Gastroenterology. 2021 Apr;160(5):1486-1501.

We agree and thank the reviewer for this comment. We have revised the manuscript and ensured that the term “microbiota-gut-brain axis” is utilized.

“The bidirectional communication between microbiota and the brain is referred to as the ‘microbiota-gut-brain axis’ (Rhee et al., 2009). The microbiota-gut-brain axis is composed of multiple pathways including the CNS, autonomic nervous system, and enteric nervous system (Cryan et al., 2019). The autonomic system includes sympathetic and parasympathetic branches consisting of afferent and efferent fibers responsible for various involuntary physiological processes (i.e., heart rate, blood pressure, digestion) (Waxenbaum et al., 2022). Afferent signals begin in the intestinal lumen and travel to the CNS through spinal, vagal, and enteric pathways, while efferent signals begin in the CNS and travel to the intestinal wall (Carabotti et al., 2015). The gut microbiota can also in-fluence the CNS through the production of several bioactive molecules such as cytokines, prostaglandins, and microbial antigens (i.e., LPS) (Farzi et al., 2018). These molecules can cross the BBB and directly influence the functioning of the CNS (Schächtle & Rosshart, 2021). Thus, both humoral and neural pathways are involved in the microbiota-gut-brain axis and can have a profound influence on brain functioning and behaviour.” (Page 11)

Reviewer 3 Report

Re: Manuscript ID: microorganisms-1952495.

This is a concise review article dealing with the role of gut dysbiosis in neurodegenerative diseases, within the intriguing microbiota-gut-brain axis during pubertal period. A particular attention was paid on possible probiotic intervention. A table or a figure would help the reader to better appreciate the content of the manuscript. Some changes are suggested to improve the paper (see PDF).

Round 2

Reviewer 1 Report

I would like to thank the authors for judiciously answering the questions and comments. I have no objections to publication of the manuscript at this point.

Author Response

Reviewer 1

General Comments

  • I would like to thank the authors for judiciously answering the questions and comments. I have no objections to publication of the manuscript at this point.

We thank the reviewer for this positive feedback.

Reviewer 2 Report

I checked the references marked with red that the author added in the version. There are five new references. Among these new references, two are old. One is from 1994, and the other is from 1991. As I suggested before, it is better to cite recent papers.

Author Response

Reviewer 2 

General Comments

  • I checked the references marked with red that the author added in the version. There are five new references. Among these new references, two are old. One is from 1994, and the other is from 1991. As I suggested before, it is better to cite recent papers

We thank the reviewer for this comment. The references from 1994 and 1991 are seminal papers on the association between tau and Alzheimer’s disease. These references were included in the manuscript under the recommendation of another reviewer.
